# MOFs/Ketjen Black-Coated Filter Paper for Spontaneous Electricity Generation from Water Evaporation

**DOI:** 10.3390/polym14173509

**Published:** 2022-08-26

**Authors:** Jingyu Li, Yexin Dai, Shipu Jiao, Xianhua Liu

**Affiliations:** School of Environmental Science and Engineering, Tianjin University, Tianjin 300354, China

**Keywords:** electricity generation, water evaporation, hybrid nanomaterials, metal-organic frameworks, salinity sensing

## Abstract

Metal-organic frameworks (MOFs) have the advantages of tunable pore sizes and porosity and have demonstrated unique advantages for various applications. This study synthesized composite MOF nanomaterials by modifying MOF801 or AlOOH with UIO66. The composite nanomaterials, UIO66/MOF801 and UIO66/AlOOH showed increased Zeta potential than their pristine form, AlOOH, UIO66 and MOF801. For the first time, the composite MOFs were used to fabricate filter paper-based evaporation-driven power generators for spontaneous electricity generation. The MOFs-KBF membrane was constructed by coating filter paper (10 × 50 mm) with composite MOFs and conductive Ketjen Black. The UIO66/MOF801 decorated device achieved a maximum open circuit voltage of 0.329 ± 0.005 V and maximum output power of 2.253 μW. The influence of salt concentration (0.1–0.5 M) on power generation was also analyzed and discussed. Finally, as a proof-of-concept application, the device was employed as a salinity sensor to realize remote monitoring of salinity. This work demonstrated the potential of flexible MOF composites for spontaneous power generation from water evaporation and provides a potential way to enhance the performance of evaporation-driven power generators.

## 1. Introduction

Modern society’s rapid development has resulted in an increasingly serious energy crisis and environmental pollution issues. Using clean and renewable energy and converting it into electrical energy is thought to be an effective way to address these issues. Many environmental energy conversion technologies have been developed for long-term power generation, including solar cells [1], thermoelectric cells [2], and moisture-driven generators [3]. However, these devices frequently require a constant supply of external energy, such as sunlight, pressure, or humidity gradients. During the conversion process, these devices always lose electrical, mechanical, and thermal energy. As a result, modern human society urgently requires a more environmentally friendly and efficient energy generation technology.

In recent years, the use of natural power of water evaporation to generate electricity [4,5,6,7,8,9] has attracted global attention due to its stable output voltage, long continuous power generation time, and no need for energy input. Electricity generation from water evaporation is produced by the capillary flow of water and the migration of charged ions. An electric double layer (EDL) is formed inside a charged capillary channel when water molecules pass through it. Some charged ions migrate with the water molecules in the EDL, resulting in a potential difference and generating current due to capillary force and water evaporation [8]. EDL is critical to the process of electricity generation from water evaporation. If the electrical properties of the charge carried by the EDL can be changed, so can the electrical properties of the counter ions entering the EDL, resulting in different voltages and currents.

Many scholars have used various materials to improve the evaporative power generation cell, mainly including carbon materials and metal oxides [10,11,12]. A transpiration-driven motor generator (TEPG) was prepared with asymmetric cotton fabric coated with carbon black [13], and the capillary flow of water produced a maximum open circuit voltage of 0.53 V and short-circuit current of 3.91 μA. Ma et al. [14] reported the synthesis of a new type of metal-organic framework (MOF) nanomaterial by growing UIO66 nanoparticles on two-dimensional AlOOH nanosheets. The evaporation-induced electricity generation battery based on the MOF nanomaterial can achieve an open circuit voltage of 1.63 ± 0.10 V. However, these previous studies also face many challenges. When a single carbon material is used as an evaporation power generation device, the output power is often lower due to its low surface charge [13,15]. MOF materials have tunable pore sizes and high porosity when compared to other materials, and they have demonstrated unique advantages in a variety of applications [16]. Currently, MOF nanomaterials can be prepared by using the hydrothermal/solvothermal method, ultrasonic method, interfacial synthesis method [17], electrochemical method, and mechanochemical synthesis method. Among them, the hydrothermal/solvothermal method is the most common and effective synthetic route to construct MOF nanomaterials. MOF composites have received extensive global attention due to their great potential to mitigate the shortcomings and/or expand the advantages of pristine MOFs. MOF composites with a variety of excellent functions can be generated after careful design, resulting in desirable properties and enhanced stability [18].

Filter paper-based devices have attracted a great deal of attention due to the special properties of paper, such as low cost, wide availability, and biodegradability. Combined with features of different electrochemical electrodes, filter paper has facilitated the rapid development of sustainable devices [19,20]. Due to its rich micro-sized pores and oxygen functional groups, a paper may be well suitable as a water-driven power generation material. A filter paper-based moisture-driven power generator has also been reported [19], and a voltage of 0.25 V and a current of 15 nA were induced on a piece of untreated print paper (1.5 cm^2^ in area).

In this study, we constructed a hydroelectric device based on MOF/Ketjen Black-coated filter paper for spontaneous electricity generation from water evaporation. The use of filter paper and Ketjen Black (KB) favors the capillary flow of water and has excellent conductivity. MOF801 and UIO66 were chosen for the fabrication of the MOF composites due to the fact that MOF801 has excellent water absorption ability while UIO66 has a relatively higher zeta potential [21,22]. Their combination is expected to create an ideal composite MOF material with both high zeta potential and good water absorption properties which is suitable for hydrovoltaic applications. The use of composite membrane based on MOF, KB, and filter paper for the evaporation-driven power generation is reported for the first time, which is believed to combine the advantages of MOF and KB, and alleviate their disadvantages. In addition, a salinity sensor was fabricated to prove the concept, providing new insights into evaporation-driven power generation and a feasible approach for designing a simple hydroelectric device for various types of applications.

## 2. Materials and Methods

### 2.1. Material

Ketjen Black (KB), sodium dodecyl benzene sulfonate (SDBS), fumaric acid, urea, terephthalic acid (H_2_DBC), zirconium oxychloride octahydrate (ZrOCl_2_·8H_2_O), zirconium tetrachloride (ZrCl_4_), N, N-dmimethylformamide (DMF, 99.8%), formic acid (FA), acetic acid, methanol, aluminum trichloride (AlCl_3_·6H_2_O), and other chemicals were purchased from Aladdin Biochemical Technology Co., Ltd. (Shanghai, China).

### 2.2. Synthesis of MOFs Materials

#### 2.2.1. Synthesis of MOF801

In 20 mL DMF and 56 mL formic acid, 1.624 g fumaric acid and 4.48 g ZrOCl_2_·8H_2_O were dissolved, respectively. These solutions were mixed and then placed in a PTFE-lined stainless-steel autoclave, which was heated in a thermostatic oven at 130 °C for 6 h to produce a white precipitate of MOF801. The precipitate was collected by filtration through a membrane filter (45 μm pore size), washed 3 times with DMF and methanol. After the cleaned material was naturally dried at room temperature, it was transferred to a vacuum oven, heated, and activated for 24 h at 15 kPa and 150 °C. Finally, the activated MOF801 was obtained as a white powder [23].

#### 2.2.2. Synthesis of UIO66

After co-dispersion of 2.1 g ZrCl4 and 1.4 g H2DBC in 10 mL DMF, 1 mL acetic acid was added. The dispersion was sonicated for 15 min before being transferred to a PTFE-lined stainless-steel autoclave and heated at 120 °C for 15 h. Following natural cooling, the obtained nanocrystals were washed 3 times with DMF and deionized water [14].

#### 2.2.3. Synthesis of AlOOH

In 10 mL deionized water, dissolve 3.36 g AlCl_3_·6H_2_O and 2.52 g urea. After sonicating for 10 min, the mixture was transferred to a PTFE-lined stainless-steel autoclave and heated in an oven at 190 °C for 10 h. After natural cooling, the product was removed from the autoclave and washed 3 times with deionized water. The obtained white product was dried and collected at room temperature [24].

#### 2.2.4. Synthesis of UIO66/AlOOH and UIO66/MOF801

0.15 g H_2_BDC, 0.45 g pre-synthesized AlOOH, and 0.2 g ZrCl_4_ were dispersed in 180 mL DMF and a mixed solvent of 20 mL H_2_O in a round bottom flask. After sonicating for 30 min, 20 mL of acetic acid was slowly added to the flask. The flask was then transferred to a heating mantle and heated under reflux at 120 °C for 12 h with constant magnetic stirring. The obtained product was washed 3 times with DMF and deionized water, dried and collected at room temperature. The preparation process of UIO66/MOF801 is similar to that of UIO66/AlOOH, with the difference being that the pre-synthesized AlOOH is replaced by MOF801 (Figure 1a).

### 2.3. Fabrication of Devices

Preparation of KB coated filter paper (KBF): 2 g KB were dispersed in 200 mL deionized water containing 4 g SDBS as surfactant. To uniformly disperse the KB, the solution was sonicated for 1 h. A piece of filter paper (10 × 50 mm) was immersed in KB suspension for 10 min to make the filter paper absorb KB fully, and then put the KB coated filter paper in an oven at 80 °C for 10 min. After drying, it was immersed in KB suspension for another 24 h, and then the KB-coated filter paper was dried in an oven at 80 °C for 30 min.

MOFs modified KBF: As shown in Figure 1b, 0.06 g MOF material (AlOOH, MOF801, UIO66, UIO66/MOF801, UIO66/AlOOH) was dispersed in 60 mL H_2_O under ultrasound. Then, a piece of KBF (10 mm × 5 mm × 0.1 mm) was put into the aqueous MOF suspension in a 100 mL autoclave and heated to 105 °C in an oven for 4 h. Finally, the MOFs-modified KB-FP (MOFs/KB-FP) was obtained by drying at 80 °C for 20 min.

Device assembly: Silver wires were used to connect the electrochemical workstation to both ends of the KBF membrane. The fixed KBF membrane was placed in a conical flask, and a peristaltic pump was used to transport 0.5 M NaCl aqueous solution into the flask (Figure 1c). The maximum open-circuit voltage and maximum short-circuit current of the KBF membrane were measured using an electrochemical workstation.

### 2.4. Characterization

Fourier transform infrared (FTIR) spectroscopy was measured within KBr slices using an FTIR spectrometer (IRAffinity-1S, Shimadzu, Kyoto, Japan) in the range of 4000–500 cm^−1^. X-ray diffraction (XRD) data were recorded on an X-ray diffraction automated diffractometer (Smartlab, Rigaku, Tokyo, Japan) using CuKa radiation from 5° to 55° at a scan rate of 5° min^−1^. Zeta potential was measured in a nanoparticle size and zeta potentiometer (Nano ZS, Malvern, Malvern City, UK). Scanning electron microscope (SEM) images and energy dispersive analysis of X-rays (EDS) were taken using a field emission scanning electron microscope (Quanta FEG 250, FEI, Hillsboro, OR, USA). Samples were fixed on the sample stage with conductive glue, sprayed with gold, and placed in a scanning electron microscope for observation. N_2_ adsorption-desorption isotherms and Brunauer–Emmett–Teller (BET) pore size distribution curves were obtained at 77 K by an ASAP 2020 HD88 system (Micromeritics, Norcross, GA, USA). The surface areas and the pore size distributions were calculated by the BET and HK method, respectively. A standard three-electrode configuration was used to measure the electrochemical performance of the power generator on an electrochemical workstation (CHI 660E, CHI Instrument Co., Ltd., Austin, TX, USA). Silver wires with a diameter of 0.5 mm were used as electrodes.

## 3. Results and Discussion

### 3.1. Material Characterization

Fourier transform infrared spectroscopy and X-ray diffraction patterns confirmed that we successfully synthesized the composite materials UIO66/MOF801 and UIO66/AlOOH. Characteristic peaks of UIO66 and MOF801 are in good agreement with literature data [25,26]. As shown in Figure 2a, the two intense bands around 1581 and 1399 cm^−1^ are associated with the O-C-O asymmetric and symmetric stretch vibrations in the carboxylate group in the ligands, respectively. In addition, no adsorption band was observed around 1700 cm^−1^, which is related to the carboxylate vibration bands of the free –COOH groups on the linkers, indicating that all carboxylates are coordinated to zirconium (Zr) ions [27,28]. At the same time, Zr-O vibrations were observed at 545, 661 and 745 cm^−1^. The strong and weak peaks at 1399 and 1508 cm^−1^ are attributed to the stretching of C-O and the aromatic organic linker C=C, respectively [25,29,30,31]. The composite material UIO66/MOF801 adds the stretching peak of aromatic organic linker C=C on the basis of the Fourier infrared spectrum of MOF801. Similarly, UIO66/AlOOH material not only has the absorption peak of -NH_2_ at 3328 cm^−1^, the absorption peaks of Al-O-H at 3091 and 1074 cm^−1^ [26,27], but also has some characteristic peaks of UIO66, such as the C=O vibration peak at 1581 cm^−1^, the aromatic organic linker C=C and C-O stretching peaks at 1399 and 1508 cm^−1^. Fourier transform infrared spectroscopy confirmed the characteristic absorption peaks of individual materials (MOF801, AlOOH and UIO66) in the synthesized UIO66/MOF801 and UIO66/AlOOH composites.

As shown in Figure 2b, the XRD spectrum of UIO66/AlOOH shows a weak and broad peak at 7.3°, corresponding to the (111) plane of UIO66 (above). It is worth mentioning that the peak intensities of UIO66/AlOOH at 14.3°, 28°, 38.3°, and 49° are all enhanced, indicating that these two-dimensional nanosheets are preferentially stacked on their base surfaces, and the modification of UIO66 does not change the original lattice shape of the AlOOH material [28]. On the contrary, UIO66/MOF801, which is modified by UIO66, only shows the weak and wide (111) plane peak of UIO66 at 7.3°. The other peaks of MOF801 are masked, and this also shows that the modification of UIO66 caused by the original lattice shape of MOF801 has changed. All the above-mentioned characterizations confirmed that UIO66 nanoparticles were successfully combined on the basis of AlOOH nanosheets and MOF801 materials, forming new nanomaterials UIO66/AlOOH and UIO66/MOF801.

The N_2_ adsorption-desorption isotherms are shown in Figure 3c. The BET surface area and total pore volume of the composite UIO66/MOF801 are 390.31 m^2^/g and 1.061 cm³/g respectively, which are significantly lower than those of pristine MOF801 (961.84 m^2^/g. and 0.261 cm^3^/g, respectively). The pore size distribution curves of MOF801 and UIO66/MOF801 (Figure 3d) demonstrate that the pore size types of the composite UIO66/MOF801 are significantly reduced. The modification of MOF801 by UIO66 was responsible for both the decrease of BET surface area and the redistribution of pore size.

It is reported that the synthesis of UIO66 with acetic acid will lead to the unsaturated coordination of the Zr_6_-cluster [32,33]. Therefore, the UIO66 synthesized in this work has a high and positive Zeta potential, which has been confirmed by Zeta potential measurement, as shown in Figure 2e. Furthermore, the composite materials UIO66/AlOOH and UIO66/MOF801 formed by the combination of UIO66 material and AlOOH and MOF801 also have a higher Zeta potential. Through comparison, it can be found that after being doped with UIO66, the Zeta potential of the raw material has been greatly improved. (i) and (ii) in Figure 2f are the structure diagrams of the core Zr_6_-cluster of UIO66 and MOF801, respectively. From the central Zr atom, UIO66 is coordinated with eight O atoms, but MOF801 is coordinated with 12 O atoms to coordinate, so the Zeta potential of MOF801 is lower than that of UIO66 [24,28].

The morphological structure and elemental composition of UIO66/AlOOH and UIO66/MOF801 on KBF were characterized by SEM and EDS mapping respectively. Figure 3a,b show the SEM images of UIO66/AlOOH-KBF with different magnifications, demonstrating that the UIO66/AlOOH nanoparticles were successively deposited on the KBF fibers. The EDS mapping images (Figure 3c–f) display that O, Al, and Zr are uniformly distributed on the membrane. The SEM images (Figure 3g,h) show that the size of UIO66/MOF801 is similar to those of UIO66/AlOOH. The EDS mapping images (Figure 3i–k) display that O and Zr are uniformly distributed on the membrane. The corresponding EDS spectra were also measured to confirm the elemental composition of the composites. Figure 3l showed signals from Al and Zr, indicating the existence of Al and Zr in the UIO66/AlOOH composite. Figure 3m showed the presence of Zr in the UIO66/MOF801 composite. At the same time, the element peaks of Na and S were also displayed. This phenomenon is due to the use of SDBS during the production of KBF. These results confirmed that the UIO66/AlOOH and UIO66/MOF801 nanocomposites were successfully synthesized.

### 3.2. Electricity Generation from Evaporation

As shown in Figure 4a, we found that the output power of KBF modified with MOFs in NaCl solution is generally 3–4 times higher than that of unmodified KBF. Among them, the voltage generated by KBF membrane in 0.5 M NaCl aqueous solution is a peak voltage V_oc_ of 0.074 ± 0.002 V. When KBF is doped with UIO66/MOF801 material with high Zeta potential and good water absorption, we found that the V_oc_ generated by KBF increased to 0.329 ± 0.004 V. However, the maximum output power of UIO66/MOF801 doped KBF is slightly lower than the MOF801 doped KBF. As shown in the EIS Nyquist diagram (Figure 4b), the semicircle diameter of UIO66/MOF801 is significantly larger than the MOF801 material. It shows that the conductivity of the composite material UIO66/MOF801 is lower than that of the MOF801 material. The electrical conductivity of the UIO66/MOF801 material limits the current output of the KBF, resulting in a drop in short-circuit current after modification of the UIO66/MOF801. When the KBF is doped with the UIO66/AlOOH, we found that the composite material UIO66/AlOOH is 2.79 times higher than the maximum output power produced by the material AlOOH. In 2020, Qinglang Ma et al. [14] used UIO66/AlOOH material to make a pure material Electricity Generation from Water Evaporation, its output voltage can reach 1.63 ± 0.10 V, but due to the weak electrical conductivity of the material, its maximum output power is only 0.15 μW. However, in this work, the maximum output power of the KB-doped UIO66/AlOOH material is 1.633 ± 0.1 μW, which is nearly 11.5 times higher than the output power of the evaporation-induced electricity generation battery constructed of pure materials. It can be hypothesized that the support’s electrical conductivity and water transport capacity, as well as the zeta potential and electrical conductivity of the modified material, are all important factors in improving the performance of the evaporation-induced generator. Table 1 compares the configurations and performance of some reported evaporative-driven generators.

### 3.3. Mechanism

Figure 5a describes the migration of ions in KBF (i) before and (ii) after modification. According to the pseudo current and electric double layer mechanism [40,44,45,46,47], as the water flow migrates in the charged pores, the oppositely charged ions will be preferentially transported to the dry end, forming a potential difference that increases or decreases along the direction of the water flow. As shown in Figure 5b(i), in the unmodified KBF membrane, the surface of the membrane pores is negatively charged. As the water flow migrates, cations preferentially migrate from the wet end to the dry end. Due to the interaction between the charges, electrons and ions migrate in the same direction, resulting in a low potential at the dry end and a high potential at the wet end. When the KBF membrane was modified with MOFs (Figure 5b(ii)), the surface of the membrane pores was positively charged. Anions are preferentially migrated to the dry end, and electrons in the KBF membrane migrate in the opposite direction to anions, resulting in a high potential at the dry end and a low potential at the wet end [48,49]. Figure 5c shows the migration of ions and electrons at the wet–dry interface in more detail.

## 4. Application

We have studied the output voltage of the UIO66/MOF801-doped evaporation induced electricity generation battery under different NaCl concentrations (0.1–0.5 M). We found that the output voltage is directly proportional to the NaCl concentration. By linearly fitting the obtained data, we obtain the linear regression equation of the NaCl concentration and the output voltage: y = 0.2123x + 0.18385 and R^2^ is 0.96, as shown in Figure 6a. As shown in Figure 6b, a wireless salinity sensing system based on evaporation power generation is built using the linear relationship between the output voltage and salinity of paper-based evaporation-driven water flow nano-liquid power generation. Appendix A details its connection diagram. The whole system is controlled by the salinity warning program in the Arduino control board. The sensing system uses the Analog Read function of the Arduino control board to read the changing voltage generated by the power generation functional area. The Arduino control board can respond to the LED lights and alarm devices to varying degrees according to the voltage change. At the same time, the Arduino control board transmits the changing voltage signal to the mobile phone through the Bluetooth serial port in real-time. In this system, when the salinity of the incoming water exceeds 0.5 M, the buzzer will alarm, and when the salinity is between 0.1–0.5 M, the LED light will flash. At the same time, the system transmits the real-time changing voltage to the mobile phone, realizing real-time monitoring and long-distance transmission of data.

## 5. Conclusions

In summary, the MOFs-KBF membrane was successfully prepared for spontaneous electricity generation by coating filter paper with MOFs together with conductive KB. We modified KBF membranes with high zeta potential composite MOF materials UIO66/MOF801 and UIO66/AlOOH. When the modified KBF membrane is used as an evaporation-driven generator, the output voltage direction is reversed, and the total output power is increased. Furthermore, we found that the generator’s performance is determined not only by the surface potential of the nanomaterial, but also by its electrical conductivity. In addition, using the linear relationship between NaCl concentration and output voltage in conjunction with an Arduino control board, we created a wireless salinity sensing system. The advantages of the KBF evaporative drive generator include its low cost, easy manufacture, and broad application. It has a high potential for harvesting ambient energy and can be used in a wide range of applications, including IoT and automation systems.

## Figures and Tables

**Figure 1 polymers-14-03509-f001:**
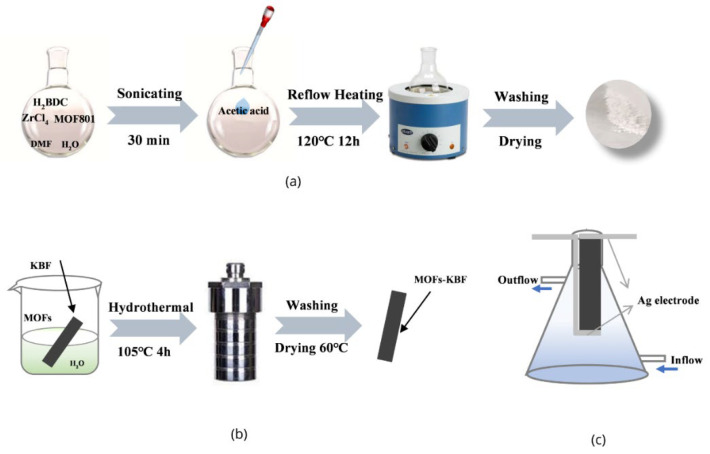
(**a**) Synthesis process diagram of UIO66/MOF801. (**b**) Process of MOFs-modified KBF. (**c**) Schematic diagram of the power generation device.

**Figure 2 polymers-14-03509-f002:**
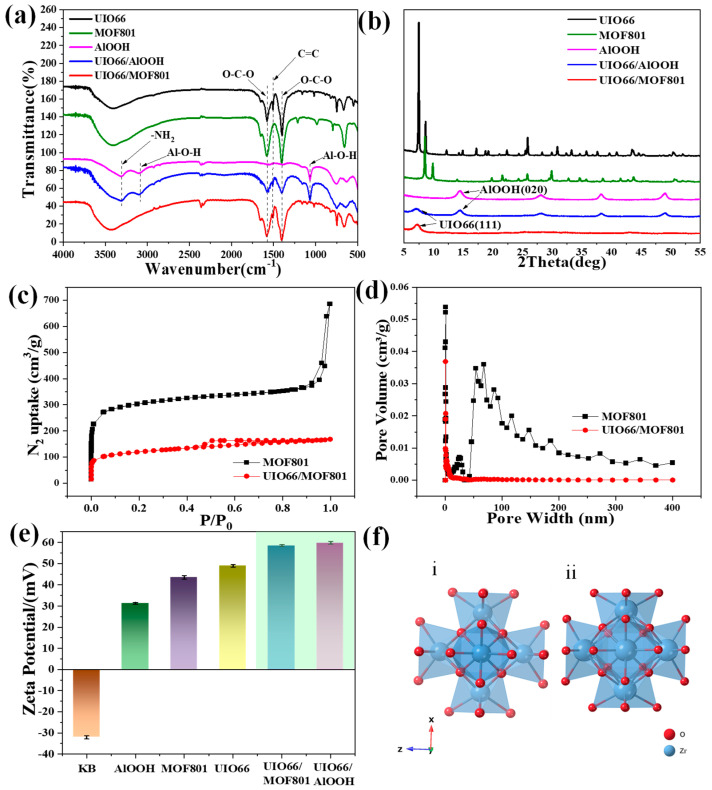
(**a**) FTIR spectra and (**b**) XRD patterns of UIO66, AlOOH, MOF801, UIO66/AlOOH and UIO66/MOF801 nanomaterials; (**c**) N_2_ adsorption-desorption isotherms and (**d**) Pore size distribution curves of MOF801 and UIO66/MOF801; (**e**) Zeta potentials of UIO66, AlOOH, MOF801, UIO66/AlOOH and UIO66/MOF801 nanomaterials; (**f**) The inner core Zr_6_-cluster structure of (i) UIO66 and (ii) MOF801. Zr, blue; O, red.

**Figure 3 polymers-14-03509-f003:**
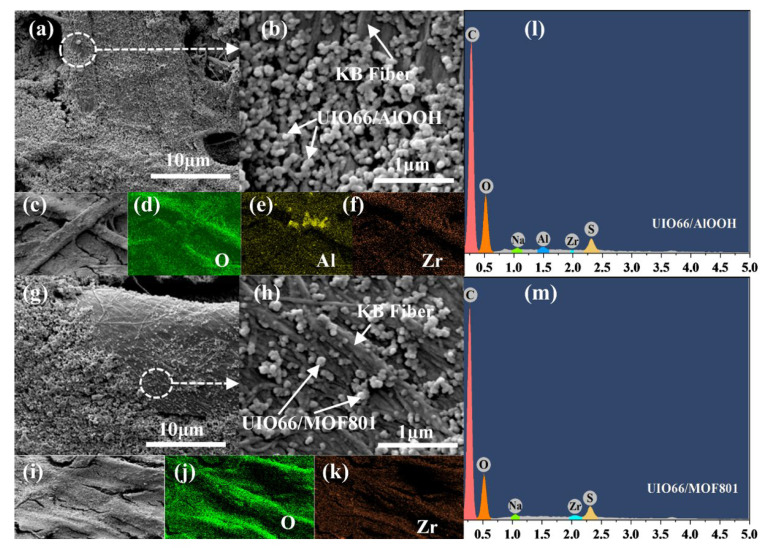
(**a**,**b**) SEM images of UIO66/AlOOH-KBF and its (**c**–**f**) EDS mapping; (**g**,**h**) SEM images of UIO66/MOF801-KBF and its (**i**–**k**) EDS mapping; EDS spectrum of (**l**) UIO66/AlOOH-KBF and (**m**) UIO66/MOF801-KBF.

**Figure 4 polymers-14-03509-f004:**
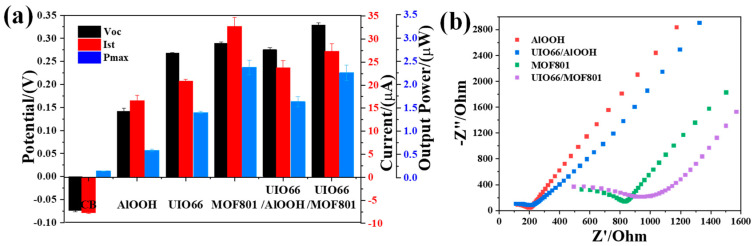
The maximum V_oc_, I_st_ and P_max_ of different MOFs modified KBF in (**a**) NaCl aqueous solution (0.5 M) and (**b**) EIS Nyquist diagram of AlOOH, UIO66/ AlOOH, MOF801 and UIO66/MOF801. The EIS was performed in 0.25 M KCl solution.

**Figure 5 polymers-14-03509-f005:**
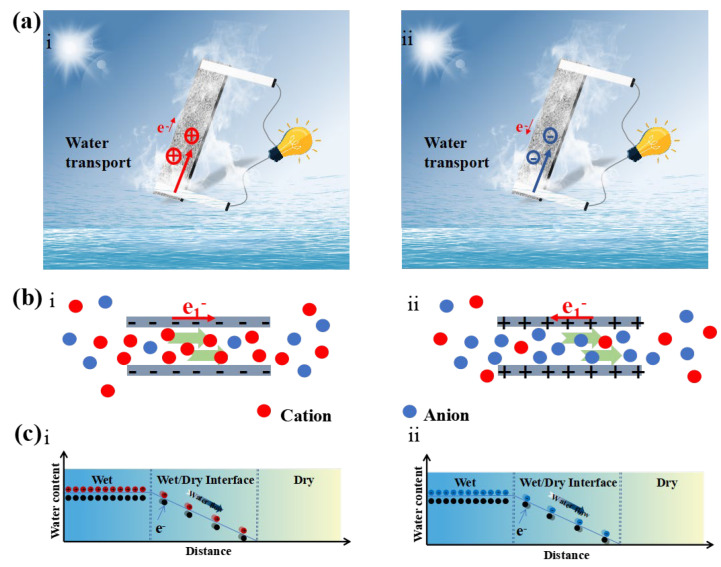
(**a**) Schematic of evaporation power generation of KBF (i) before and (ii) after modification; (**b**) Schematic of the ion-selective transport through a nanochannel before and (ii) after modification; (**c**) Mechanism of the current generation before (i) and after (ii) modification.

**Figure 6 polymers-14-03509-f006:**
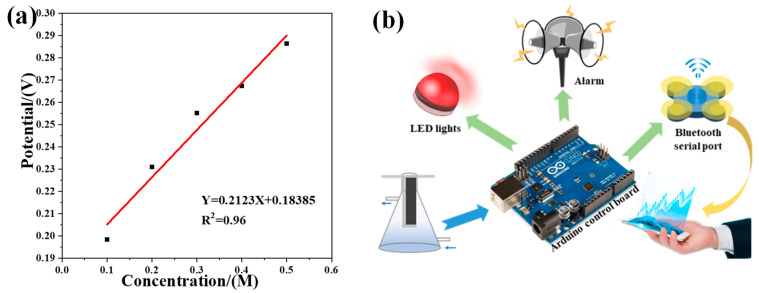
(**a**) The linear relationship between concentration and output voltage; (**b**) Schematic of the sensing system.

**Table 1 polymers-14-03509-t001:** Configuration and performance comparison of various evaporation-driven power generators.

Active Material	Solution	V_oc_ (V)	I_st_ (μA)	Refs
UIO66/MOF801—KBF	0.5 M NaCl	0.329 ± 0.005	27.37 ± 1.63	This work
UIO66/AlOOH—KBF	0.5 M NaCl	0.275 ± 0.004	23.72 ± 1.61	This work
MOF801—KBF	0.5 M NaCl	0.289 ± 0.003	32.77 ± 1.92	This work
UIO66—KBF	0.5 M NaCl	0.268 ± 0.007	20.90 ± 1.14	This work
CNPs	Water	0.89	0.38	[34]
GO+PAAS	Water	0.6	-	[35]
SS–PVA	Water	0.6	2	[36]
GO	Water	1.5	0.136	[37]
AlOOH/UIO66	Water	1.63 ± 0.10	0.49	[14]
Ni-Al layered double hydroxide	Water	0.6	0.3	[38]
Citric acid-modified wood	Water	0.3	10	[39]
ZnO porous film	Water	0.4	0.02	[40]
Silicon nanowire array	Water	0.4	55 μA cm^−2^	[41]
Carbon black	Water	0.005	-	[42]
Graphene/carbon cloth	0.5 M NaCl	0.37	-	[43]
Print paper	Water	0.25	0.015	[19]

## Data Availability

Not applicable.

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
