# Peer review of "MOFs/Ketjen Black-Coated Filter Paper for Spontaneous Electricity Generation from Water Evaporation"

_polymers, 2022, doi:10.3390/polym14173509_

Round 1

Reviewer 1 Report

Dear Authors

Congratulations for your work and the presentation of newly designed material properties in the paper.

The experimental setup is clearly presented.

The unusual result that draw my attention was the IR spectra presented in figure 2a. It seems like the UIO66 has the spectra of terephtalic acid without the band from ~ 1710 cm-1, and MOF801 has the spectra of fumaric acid without the same specific band for C=O.

You presented the 1581 cm-1 as being related to C=O vibration, which is unusual low. 

Please explain this result and add references to support your explanation.

The phrase " By comparing the Fourier transform 164 infrared spectra before and after composite, UIO66 successfully composited with MOF801 165 and AlOOH, forming composite materials UIO66/MOF801 and UIO66/AlOOH with new 166 structures." (page 4, lines 164-167) should be reformulated. You should give some additional information related to IR spectra to support this affirmation.

Author Response

Dear Reviewer,

Thank you very much for your comments on our manuscript “MOFs/Ketjen Black -Coated Filter Paper for Spontaneous Electricity Generation from Water Evaporation”. (Manuscript ID: polymers-1883621). These comments are all valuable and very helpful for revising and improving our manuscript, as well as the important guiding significance to our researches. We have studied the comments carefully and have made revisions as the reviewers suggested. Detail revisions are marked with red color in the revised manuscript.

Response to Reviewer 1

  1. The unusual result that draw my attention was the IR spectra presented in figure 2a. It seems like the UIO66 has the spectra of terephtalic acid without the band from ~ 1710 cm-1, and MOF801 has the spectra of fumaric acid without the same specific band for C=O. You presented the 1581 cm-1 as being related to C=O vibration, which is unusual low. Please explain this result and add references to support your explanation.

Response: Thank you for the suggestion. There is a coordination effect between metal ions and organic ligands during the formation of MOFs. In the revised manuscript, we have added the explanation and references involved.

“Characteristic peaks of UIO66 and MOF801 are in good agreement with literature data [25, 26]. As shown in Figure 2a, the two intense bands around 1581 and 1399 cm−1 are associated with the O-C-O asymmetric and symmetric stretch vibrations in the carboxylate group in the ligands, respectively. In addition, no adsorption band was observed around 1700 cm-1, which is related to the carboxylate vibration bands of the free –COOH groups on the linkers, indicating that all carboxylates are coordinated to zirconium(Zr) ions [27, 28]. At the same time, Zr-O vibrations were observed at 545, 661 and 745 cm-1. The strong and weak peaks at 1399 and 1508 cm-1 are attributed to the stretching of C-O and the aromatic organic linker C=C, respectively [29-32].”

  • Pang, S.; Wu, Y.; Zhang, X.; Li, B.; Ouyang, J.; Ding, M., Immobilization of laccase via adsorption onto bimodal mesoporous Zr-MOF. Process Biochem 2016, 51, 229-239. https://doi.org/https://doi.org/10.1016/j.procbio.2015.11.033.
  • Zuhra, Z.; Zhao, Z.; Qin, L.; Zhou, Y.; Zhang, L.; Ali, S.; Tang, F.; Ping, E., In situ formation of a multiporous MOF(Al)@γ-AlOOH composite material: A versatile adsorbent for both N- and S-heterocyclic fuel contaminants with high selectivity. Eng. J. 2019, 360, 1623-1632. https://doi.org/https://doi.org/10.1016/j.cej.2018.10.205.
  • Comotti, A.; Bracco, S.; Sozzani, P.; Horike, S.; Matsuda, R.; Chen, J.; Takata, M.; Kubota, Y.; Kitagawa, S., Nanochannels of two distinct cross-sections in a porous Al-based coordination polymer. Am. Chem. Soc. 2008, 130, 13664-13672. https://doi.org/10.1021/ja802589u.
  • Nan, J.; Dong, X.; Wang, W.; Jin, W., Formation mechanism of metal–organic framework membranes derived from reactive seeding approach. Microporous Mesoporous Mater. 2012, 155, 90-98. https://doi.org/https://doi.org/10.1016/j.micromeso.2012.01.010.
  • Farzaneh, F.; Kabir, N.; Geravand, E.; Ghiasi, M.; Ghandi, M., Immobilization and DFT studies of Tin chloride on UiO-66 metal–organic frameworks as active catalyst for enamination of acetylacetone. Iran. Chem. Soc. 2019, 16, 2231-2241. https://doi.org/10.1007/s13738-019-01693-4.
  • Zhao, F.; Cheng, H. H.; Zhang, Z. P.; Jiang, L.; Qu, L. T., Direct Power Generation from a Graphene Oxide Film under Moisture. Mater. 2015, 27, 4351-4357. https://doi.org/10.1002/adma.201501867.
  • Abid, H. R.; Tian, H.; Ang, H.-M.; Tade, M. O.; Buckley, C. E.; Wang, S., Nanosize Zr-metal organic framework (UiO-66) for hydrogen and carbon dioxide storage. Eng. J. 2012, 187, 415-420. https://doi.org/https://doi.org/10.1016/j.cej.2012.01.104.
  • Pang, S.; Wu, Y.; Zhang, X.; Li, B.; Ouyang, J.; Ding, M., Immobilization of laccase via adsorption onto bimodal mesoporous Zr-MOF. Process Biochem 2016, 51, 229-239. https://doi.org/https://doi.org/10.1016/j.procbio.2015.11.033.
  1. The phrase " By comparing the Fourier transform 164 infrared spectra before and after composite, UIO66 successfully composited with MOF801 165 and AlOOH, forming composite materials UIO66/MOF801 and UIO66/AlOOH with new 166 structures." (page 4, lines 164-167) should be reformulated. You should give some additional information related to IR spectra to support this affirmation.

Response: Thank you for the suggestion. In the revised manuscript, we have revised the above sentence as follows.

“Fourier transform infrared spectroscopy confirmed the characteristic absorption peaks of individual materials (MOF801, AlOOH and UIO66) in the synthesized UIO66/MOF801 and UIO66/AlOOH composites.”

We have tried our best to improve the manuscript and made the corresponding corrections according to the comments. These corrections have been marked in the revised manuscript. We appreciate for your warm work earnestly, and hope that the corrections will meet with approval. Once again, thank you very much for your comments and suggestions.

Best regards

Dr. Prof. Xianhua Liu

School of Environmental Science & Engineering

Tianjin University

Reviewer 2 Report

In this manuscript, the authors prepared UIO66/MOF801, UIO66/ALOOH composite MOF nanomaterials and the resulting composite MOFs were used to fabricate filter paper-based evaporation-driven power generators for the first time. The manuscript is well-written. I have some minor suggestions:

1. In this manuscript, composite MOF nanomaterials were prepared. Accordingly, more background information about MOF nanomaterial and composite MOF material should be given in the introduction section. For example, What are the general methods for the MOF nanomaterial preparation and how the MOF nanomaterial is superior to the bulk MOF material. Some papers can be cited: "Templated interfacial synthesis of metal-organic framework (MOF) nano-and micro-structures with precisely controlled shapes and sizes." Communications Chemistry 4.1 (2021): 1-10.; "Metal–organic framework nanoparticles." Advanced Materials 30.37 (2018): 1800202.; "Nanoarchitectured design of porous materials and nanocomposites from metal-organic frameworks." Advanced materials 29.12 (2017): 1604898.

In addition, why MOF801 and UIO66 were chosen for the current study? The reason should be specified.

2. How SEM samples were prepared? Were samples coated by carbon or gold? If yes, detailed experimental procedures should be given in the Characterization section. 

3. Figure 3, I suggest the authors to also include the SEM images of  pure KBF fibers for a better comparison. 

4. Table 1, please include error bars in the table for the current study.

5. High porosity and large surface area are the most desirable characteristics of MOFs. Have the authors tested the porosity and  surface area of the composite MOF material prepared in the current study? 

Author Response

Dear Reviewer,

Thank you very much for your comments on our manuscript “MOFs/Ketjen Black -Coated Filter Paper for Spontaneous Electricity Generation from Water Evaporation”. (Manuscript ID: polymers-1883621). These comments are all valuable and very helpful for revising and improving our manuscript, as well as the important guiding significance to our researches. We have studied the comments carefully and have made revisions as the reviewers suggested. Detail revisions are marked with red color in the revised manuscript.

Response to Reviewer 2

In this manuscript, the authors prepared UIO66/MOF801, UIO66/ALOOH composite MOF nanomaterials and the resulting composite MOFs were used to fabricate filter paper-based evaporation-driven power generators for the first time. The manuscript is well-written. I have some minor suggestions:

  1. In this manuscript, composite MOF nanomaterials were prepared. Accordingly, more background information about MOF nanomaterial and composite MOF material should be given in the introduction section. For example, What are the general methods for the MOF nanomaterial preparation and how the MOF nanomaterial is superior to the bulk MOF material. Some papers can be cited: "Templated interfacial synthesis of metal-organic framework (MOF) nano-and micro-structures with precisely controlled shapes and sizes." Communications Chemistry 4.1 (2021): 1-10.; "Metal–organic framework nanoparticles." Advanced Materials 30.37 (2018): 1800202.; "Nanoarchitectured design of porous materials and nanocomposites from metal-organic frameworks." Advanced materials 29.12 (2017): 1604898.

Response: Thank you for the suggestion. We agree with the reviewers that the references and perspectives provided are a great help in supporting the thesis's point of view. In the revised manuscript, we have added the following sentences and cited the above references.

“MOF materials have tunable pore sizes and high porosity when compared to other materials, and they have demonstrated unique advantages in a variety of applications [16]. Currently, MOF nanomaterials can be prepared by using hydrothermal/solvothermal method, ultrasonic method, interfacial synthesis method [17], electrochemical method, and mechanochemical synthesis method. Among them, the hydrothermal/solvothermal method is the most common and effective synthetic routes to fabricate MOF nanomaterials. MOF composites have received extensive global attention due to their great potentials to mitigate the shortcomings or expand the advantages of pristine MOFs. MOF composites with a variety of excellent functions can be generated after careful design, resulting in desirable properties and enhanced stability [18].”

  1. Kaneti, Y. V.; Tang, J.; Salunkhe, R. R.; Jiang, X. C.; Yu, A. B.; Wu, K. C. W.; Yamauchi, Y., Nanoarchitectured Design of Porous Materials and Nanocomposites from Metal-Organic Frameworks. Mater. 2017, 29. https://doi.org/10.1002/adma.201604898.
  2. Meng, L. Y.; Yu, B. Y.; Qin, Y., Templated interfacial synthesis of metal-organic framework (MOF) nano- and micro-structures with precisely controlled shapes and sizes. Chem. 2021, 4. https://doi.org/10.1038/s42004-021-00522-1.
  3. Wang, S. Z.; McGuirk, C. M.; d'Aquino, A.; Mason, J. A.; Mirkin, C. A., Metal-Organic Framework Nanoparticles. Mater. 2018, 30. https://doi.org/10.1002/adma.201800202.

In addition, why MOF801 and UIO66 were chosen for the current study? The reason should be specified.

Response: Thank you for the suggestion. We have added the reasons for choosing MOF801 and UIO66 for compounding. In the revised manuscript, we added the following sentences:

“MOF801 and UIO66 were chosen for fabrication the MOF composites due to the fact that MOF801 has excellent water absorption ability and UIO66 has a relatively higher zeta potential [21, 22]. It is believed that their combination can create an ideal composite MOF material with both high zeta potential and good water absorption properties which is suitable for hydrovoltaic applications.”

  1. Li, J.; Liu, K.; Ding, T. P.; Yang, P. H.; Duan, J. J.; Zhou, J., Surface functional modification boosts the output of an evaporation-driven water flow nanogenerator. Nano Energy 2019, 58, 797-802. https://doi.org/10.1016/j.nanoen.2019.02.011.
  2. Chueh, C.-C.; Chen, C.-I.; Su, Y.-A.; Konnerth, H.; Gu, Y.-J.; Kung, C.-W.; Wu, K. C. W., Harnessing MOF materials in photovoltaic devices: recent advances, challenges, and perspectives. Mater. Chem. A 2019, 7, 17079-17095. https://doi.org/10.1039/c9ta03595h.
  3. How SEM samples were prepared? Were samples coated by carbon or gold? If yes, detailed experimental procedures should be given in the Characterization section.

Response: Thank you for the comment. We have added detailed preparation procedures for SEM samples in the characterization section in the main text.

“Scanning electron microscope (SEM) images and energy dispersive analysis of X-rays (EDS) were taken using a field emission scanning electron microscope (Quanta FEG 250, FEI, USA). Samples were fixed on the sample stage with conductive glue, sprayed with gold, and placed in a scanning electron microscope for observation.”

  1. Figure 3, I suggest the authors to also include the SEM images of pure KBF fibers for a better comparison.

Response: Thank you for the suggestion. We agree with the reviewer's suggestion to do SEM images of pure KBF for better comparison. However, we cannot conduct the SEM measurement due to the long reservation list for the SEM instrument in the analyses center of our university during Covid-19 pandemic.

  1. Table 1, please include error bars in the table for the current study.

Response: Thank you for the suggestion. We have added error bars for the current study in Table 1.

Table 1. Configuration and performance comparison of various evaporation-driven power generators.

Active material

Solution

Voc (V)

Ist (μA)

Refs

UIO66/MOF801 - KBF

0.5 M NaCl

0.329±0.005

27.37±1.63

This work

UIO66/AlOOH - KBF

0.5 M NaCl

0.275±0.004

23.72±1.61

This work

MOF801 - KBF

0.5 M NaCl

0.289±0.003

32.77±1.92

This work

UIO66 - KBF

0.5 M NaCl

0.268±0.007

20.90±1.14

This work

CNPs

Water

0.89

0.38

[34]

GO+PAAS

Water

0.6

-

[35]

SS–PVA

Water

0.6

2

[36]

GO

Water

1.5

0.136

[37]

AlOOH/UIO66

Water

1.63±0.10

0.49

[14]

Ni-Al layered double hydroxide

Water

0.6

0.3

[38]

Citric acid-modified wood

Water

0.3

10

[39]

ZnO porous film

Water

0.4

0.02

[40]

Silicon nanowire array

Water

0.4

55 μA cm-2

[41]

Carbon black

Water

0.005

-

[42]

Graphene/carbon cloth

0.5 M NaCl

0.37

-

[43]

Print paper

Water

0.25

0.015

[19]

  1. Ma, Q. L.; He, Q. Y.; Yin, P. F.; Cheng, H. F.; Cui, X. Y.; Yun, Q. B.; Zhang, H., Rational Design of MOF-Based Hybrid Nanomaterials for Directly Harvesting Electric Energy from Water Evaporation. Mater. 2020, 32. https://doi.org/10.1002/adma.202003720.
  2. Gao, X.; Xu, T.; Shao, C. X.; Han, Y. Y.; Lu, B.; Zhang, Z. P.; Qu, L. T., Electric power generation using paper materials. Mater. Chem. A 2019, 7, 20574-20578. https://doi.org/10.1039/c9ta08264f.
  3. Liu, K.; Ding, T. P.; Li, J.; Chen, Q.; Xue, G. B.; Yang, P. H.; Xu, M.; Wang, Z. L.; Zhou, J., Thermal-Electric Nanogenerator Based on the Electrokinetic Effect in Porous Carbon Film. Energy Mater. 2018, 8. https://doi.org/10.1002/aenm.201702481.
  4. Huang, Y. X.; Cheng, H. H.; Yang, C.; Yao, H. Z.; Li, C.; Qu, L. T., All-region-applicable, continuous power supply of graphene oxide composite. Energy Environ. Sci. 2019, 12, 1848-1856. https://doi.org/10.1039/c9ee00838a.
  5. Wang, H. Y.; Cheng, H. H.; Huang, Y. X.; Yang, C.; Wang, D. B.; Li, C.; Qu, L. T., Transparent, self-healing, arbitrary tailorable moist-electric film generator. Nano Energy 2020, 67. https://doi.org/10.1016/j.nanoen.2019.104238.
  6. Huang, Y. X.; Cheng, H. H.; Yang, C.; Zhang, P. P.; Liao, Q. H.; Yao, H. Z.; Shi, G. Q.; Qu, L. T., Interface-mediated hygroelectric generator with an output voltage approaching 1.5 volts. Commun. 2018, 9. https://doi.org/10.1038/s41467-018-06633-z.
  7. Tian, J. L.; Zang, Y. H.; Sun, J. C.; Qu, J. Y.; Gao, F.; Liang, G. Y., Surface charge density-dependent performance of Ni-Al layered double hydroxide-based flexible self-powered generators driven by natural water evaporation. Nano Energy 2020, 70. https://doi.org/10.1016/j.nanoen.2020.104502.
  8. Zhou, X. B.; Zhang, W. L.; Zhang, C. L.; Tan, Y.; Guo, J. C.; Sun, Z. N.; Deng, X., Harvesting Electricity from Water Evaporation through Microchannels of Natural Wood. ACS Appl. Mater. Interfaces 2020, 12, 11232-11239. https://doi.org/10.1021/acsami.9b23380.
  9. Yoon, S. G.; Yang, Y.; Yoo, J.; Jin, H.; Lee, W. H.; Park, J.; Kim, Y. S., Natural Evaporation-Driven Ionovoltaic Electricity Generation. ACS Appl. Electron. Mater. 2019, 1, 1746-1751. https://doi.org/10.1021/acsaelm.9b00419.
  10. Qin, Y.; Wang, Y.; Sun, X.; Li, Y.; Xu, H.; Tan, Y.; Li, Y.; Song, T.; Sun, B., Constant Electricity Generation in Nanostructured Silicon by Evaporation-Driven Water Flow. Chem. Int. Ed. 2020, 59, 10619-10625. https://doi.org/10.1002/anie.202002762.
  11. Hou, B. F.; Cui, Z. Q.; Zhu, X.; Liu, X. H.; Wang, G.; Wang, J. Y.; Mei, T.; Li, J. H.; Wang, X. B., Functionalized carbon materials for efficient solar steam and electricity generation. Chem. Phys. 2019, 222, 159-164. https://doi.org/10.1016/j.matchemphys.2018.10.006.
  12. Hou, B.; Kong, D.; Qian, J.; Yu, Y.; Cui, Z.; Liu, X.; Wang, J.; Mei, T.; Li, J.; Wang, X., Flexible and portable graphene on carbon cloth as a power generator for electricity generation. Carbon 2018, 140, 488-493. https://doi.org/10.1016/j.carbon.2018.09.005.
  13. High porosity and large surface area are the most desirable characteristics of MOFs. Have the authors tested the porosity and surface area of the composite MOF material prepared in the current study?

Response: Thank you for your suggestion. We have added porosity and surface area tests to MOF801 and UIO66/MOF801. We add N2 adsorption-desorption isotherms and pore size distribution curves (Figure 2c and 2d). We also add the following sentences to the main text to show the changes in porosity and surface area of MOF materials before and after compounding.

“The N2 adsorption-desorption isotherms are shown in Figure 3c. The BET surface area and total pore volume of the composite UIO66/MOF801 are 390.31 m2/g and 1.061 cm³/g respectively, which are significantly lower than those of pristine MOF801 (961.84 m2/g. and 0.261 cm3/g, respectively). The pore size distribution curves of MOF801 and UIO66/MOF801 (Figure 3d) demonstrate that the pore size types of the composite UIO66/MOF801 are significantly reduced. The modification of MOF801 by UIO66 was responsible for both the decrease of BET surface area and the redistribution of pore size.”

Figure 2. (a) FTIR spectra and (b) XRD patterns of UIO66, AlOOH, MOF801, UIO66/AlOOH and UIO66/MOF801 nanomaterials; (c) N2 adsorption-desorption isotherms and (d) Pore size distribution curves of MOF801 and UIO66/MOF801; (e) Zeta potentials of UIO66, AlOOH, MOF801, UIO66/AlOOH and UIO66/MOF801 nanomaterials; (f) The inner core Zr6-cluster structure of (i)UIO66 and (ii) MOF801. Zr, blue; O, red.

We have tried our best to improve the manuscript and made the corresponding corrections according to the comments. These corrections have been marked in the revised manuscript. We appreciate for your warm work earnestly, and hope that the corrections will meet with approval. Once again, thank you very much for your comments and suggestions.

Best regards

Dr. Prof. Xianhua Liu

School of Environmental Science & Engineering

Tianjin University
